# Transient Receptor Potential Ankyrin 1 Ion Channel Is Expressed in Osteosarcoma and Its Activation Reduces Viability

**DOI:** 10.3390/ijms25073760

**Published:** 2024-03-28

**Authors:** Lina Hudhud, Katalin Rozmer, Angéla Kecskés, Krisztina Pohóczky, Noémi Bencze, Krisztina Buzás, Éva Szőke, Zsuzsanna Helyes

**Affiliations:** 1Department of Pharmacology and Pharmacotherapy, Center for Neuroscience, Medical School, University of Pécs, 7624 Pécs, Hungarykovacs-rozmer.katalin@pte.hu (K.R.); angela.kecskes@aok.pte.hu (A.K.); pohoczkykriszti@gmail.com (K.P.); bencze.noemi89@gmail.com (N.B.); eva.szoke@aok.pte.hu (É.S.); 2National Laboratory for Drug Research and Development, 1077 Budapest, Hungary; 3Department of Nursing, Faculty of Medicine, Umeå University, 901 87 Umeå, Sweden; 4Department of Pharmaceutical Chemistry, University of Pécs, 7624 Pécs, Hungary; 5Hungarian Research Network, Chronic Pain Research Group, 7624 Pécs, Hungary; 6Department of Pharmacology, Faculty of Pharmacy, University of Pécs, 7624 Pécs, Hungary; 7Department of Immunology, Albert Szent-Györgyi Medical School, Faculty of Science and Informatics, University of Szeged, 6720 Szeged, Hungary; buzas.krisztina@brc.hu; 8Laboratory of Microscopic Image Analysis and Machine Learning, Institute of Biochemistry, Biological Research Centre, Eötvös Loránd Research Network (ELKH), 6726 Szeged, Hungary; 9PharmInVivo Ltd., 7629 Pécs, Hungary

**Keywords:** osteosarcoma, TRPA1, TRPV1, mustard oil, capsaicin, RNAscope in situ hybridization, radioactive _45_Ca^2+^ uptake, cell viability

## Abstract

Osteosarcoma is a highly malignant, painful cancer with poor treatment opportunities and a bad prognosis. Transient receptor potential ankyrin 1 (TRPA1) and vanilloid 1 (TRPV1) receptors are non-selective cation channels that have been of great interest in cancer, as their expression is increased in some malignancies. In our study we aim to characterize the expression and functionality of the TRPA1 and TRPV1 channels in human and mouse osteosarcoma tissues and in a mouse cell line. *TRPA1/Trpa1* and *TRPV1/Trpv1* mRNA expressions were demonstrated by PCR gel electrophoresis and RNAscope in situ hybridization. The function of these channels was confirmed by their radioactive _45_Ca^2+^ uptake in response to the TRPA1 agonist, Allyl-isothiocyanate (AITC), and TRPV1 agonist, capsaicin, in K7M2 cells. An ATP-based K2M7 cell viability luminescence assay was used to determine cell viability after AITC or capsaicin treatments. Both *TRPA1/Trpa1* and *TRPV1/Trpv1* were expressed similarly in human and mouse osteosarcoma tissues, while *Trpa1* transcripts were more abundantly present in K7M2 cells. TRPA1 activation with 200 µM AITC induced a significant _45_Ca^2+^ influx into K7M2 cells, and the antagonist attenuated this effect. In accordance with the lower *Trpv1* expression, capsaicin induced a moderate _45_Ca^2+^ uptake, which did not reach the level of statistical significance. Both AITC and capsaicin significantly reduced K7M2 cell viability, demonstrating EC_50_ values of 22 µM and 74 µM. The viability-decreasing effect of AITC was significantly but only partially antagonized by HC-030031, but the action of capsaicin was not affected by the TRPV1 antagonist capsazepine. We provide here the first data on the functional expression of the TRPA1 and TRPV1 ion channels in osteosarcoma, suggesting novel diagnostic and/or therapeutic perspectives.

## 1. Introduction

Osteosarcoma (OS) is the most common and painful malignant bone tumor derived from mesenchymal stem cells and usually develops at the metaphysis of long bones, especially the distal femur and proximal tibia [1,2]. Current therapeutic strategies, including surgical excision, chemotherapy, and radiotherapy, have dramatically improved the prognosis of patients with localized OS, with a 5-year survival rate of 60–78%. However, with a high ability for local invasion and early lung metastasis, OS is often fatal in both children and adults, with an expected 5-year survival of only 20–30% [3]. Despite the treatment options, there is an urgent need for novel effective therapies to control the proliferation and migration of OS cells, particularly those with metastatic potential.

Transient Receptor Potential (TRP) receptors are non-selective cation channels which induce a Ca^2+^ influx in response to their activation and consequently trigger a range of signaling pathways regulating cell proliferation, survival, and homeostasis [4]. Therefore, they play crucial roles in malignant transformation, migration, and apoptosis [5]. TRP ankyrin 1 (TRPA1) is activated endogenously by substances produced at sites of tissue damage and inflammation, including reactive oxygen, nitrogen, and carbonyl species. This channel can also be activated by exogenous compounds such as cinnamaldehyde, allyl isothiocyanate (AITC, mustard oil), and allicin, which serve as experimental tools for functional investigations [6]. TRPA1 is highly permeable to Ca^2+^, which in turn affects cancer cells’ growth and migration and inhibits apoptosis in cancer cells. The Cancer Genome Atlas data report that a high expression of the TRPA1 gene is associated with improved survival in intrahepatic biliary duct and bladder as well as liver cancers. TRPA1 modulates cellular processes and inhibits pancreatic ductal adenocarcinoma Panc-1 cell migration through both pore-dependent and independent mechanisms [7]. Furthermore, a study employing lipoyl-based TRPA1 antagonists demonstrated their ability to decrease the migration of osteosarcoma cells and reduce the expression of pro-inflammatory cytokines [8]. The structurally and functionally similar TRP vanilloid 1 (TRPV1) channel is best known for its ability to be activated by the pungent compound of chili pepper, capsaicin. It can also be activated by other physical and chemical stimuli, such as temperatures over 43 °C, protons (pH < 6), and vanilloids. TRPV1 activation also induces a Ca^2+^ influx, which in excess can lead to cell death. A higher expression of TRPV1 was reported in human primary brain cancers [9], chronic pancreatitis and pancreatic cancer [10], and breast [11] and thyroid cancers [12]. Different TRPV1 agonists or antagonists had effects on cancer cell proliferation. There are reports on the ability of capsaicin to induce cell death in urothelial cancer and glioma through TRPV1-dependent excessive Ca^2+^ influxes [13,14]. However, capsaicin was also shown to reduce proliferation through TRPV1-independent mechanisms, such as in the human OS MG63 cell line, where it decreased viability both via TRPV1-dependent and -independent pathways [15].

Recent drug discovery studies revealed the TRPA1 and TRPV1 channels to be potential therapeutic targets in many diseases including asthma, pain, neuro-psychiatric disorders, and different types of cancers. It has been shown that the expression of these TRP channels is associated with tumor progression in prostate, bladder, pancreas, and colon cancers [16], as well as melanoma and glioma [17].

Although multiple studies have characterized the expression and role of the TRPA1 and TRPV1 channels in tumors, there are no sufficient data regarding OS. Therefore, here we have investigated the functional expression of TRPA1 and TRPV1 in human and mouse OS tissues, as well as in K7M2 cells.

## 2. Results

### 2.1. TRPA1/Trpa1 and TRPV1/Trpv1 mRNAs Are Expressed in Human and Mouse OS Tissues and in K7M2 Cells

The *TRPA1* and *TRPV1* RNAscope in situ hybridization showed the expression of *TRPA1* and *TRPV1* mRNA as being approximately at the same level in two different human OS samples, characterized by an osteoid mass with hematoxylin–eosin (H&E) staining (Figure 1A,D). The mRNA transcripts of both channels were present on ezrin-positive cells, demonstrating that they are OS cells (Figure 1B,C,E,F).

In the case of mouse OS samples, seen as a similar osteoid mass (Figure 2C), apparently the same level of expression was observed for both *Trpa1* and *Trpv1* in tumor cells (Figure 2D,E). In the control, healthy bone tissue of the mouse, supported by H&E staining, few transcripts corresponding to both receptors can be seen (Figure 2A,B).

Both the PCR gel electrophoresis and RNAscope in situ hybridization showed abundant *Trpa1* but low *Trpv1* expression in the mouse K7M2 OS cell line, as presented in Figure 3A,B.

### 2.2. The TRPA1 Agonist AITC Induces Radioactive _45_Ca^2+^ Uptake in K7M2 Cells

The TRPA1 agonist AITC (10, 100, and 200 µM) induced a concentration-dependent _45_Ca^2+^ influx of 400 ± 151, 470 ± 71, and 1037 ± 220 CPM into K7M2 cells, respectively, reaching the level of statistical significance at its highest concentration. The TRPA1 antagonist HC-030031 (10 µM) significantly reduced AITC-induced _45_Ca^2+^ uptake (255 ± 105 CPM, Figure 4A). The TRPV1 agonist capsaicin, at 100 nm and 1 µM concentrations, did not induce a significant _45_Ca^2+^ uptake, and the radioactive signal was not influenced by the TRPV1 antagonist capsazepine (Figure 4B).

### 2.3. TRPA1 and TRPV1 Agonists Reduce K7M2 Cell Viability

The incubation of K7M2 cells with five different concentrations of AITC for 48 h resulted in significant concentration-dependent K7M2 cell viability decreases ranging from 74.76% ± 2.767 to 0.2% ± 0.045 at 5–200 µM, respectively, with an EC_50_ of 22 µM ± 1.08 (Figure 5A,E). The capsaicin treatment at 20, 50, 100, and 200 µM for 48 h led to significant cell death (viability of 79.42% ± 3.731, 65.29% ± 3.515, 45.55% ± 2.188, and 12.46 ± 0.324, respectively) compared to vehicle control with an EC_50_ of 74 µM ± 1.07. (Figure 5C,E). The vehicle dimethyl sulfoxide (DMSO) also decreased K7M2 cell viability at its highest concentrations (Figure 5A,C); therefore, our results are compared to their respective controls.

The treatment of cells with 10 µM of TRPA1 antagonist HC-030031 did not affect K7M2 cell viability (Figure 5A), while 10 µM of the TRPV1 antagonist capsazepine significantly decreased K7M2 cell viability (Figure 5C).

Although an HC-030031 (10 μM) pre-treatment for 10 min statistically significantly diminished the cell death induced by the EC_50_ concentration (22 μM) of AITC, this effect was only minimal and did not seem to be biologically relevant (Figure 5B). In contrast, the capsazepine (10 μM) pretreatment did not reverse the effect of capsaicin at the 74 µM EC_50_ concentration (Figure 5D).

Both AITC and capsaicin demonstrated similar effects on the viability of Chinese hamster ovary (CHO) cells not expressing TRPA1 and TRPV1 to those observed on K7M2 cells, However, the viability-decreasing effect of AITC was not as statistically significant as in the K7M2 cells (Figure 5F).

## 3. Discussion

We provide here the first evidence of functional TRPA1 expression in human and mouse OS and the potential ability of its activation to reduce cancer cell viability. Furthermore, our TRPV1 expression data support earlier findings showing the cytotoxic effects of capsaicin on human OS cells [14].

Both the TRPA1 and TRPV1 channels were originally described in thinly myelinated Aδ- and unmyelinated C sensory fibers, where they mediate pain and neurogenic inflammation [18]. Therefore, most studies have examined their effects as being nociceptor and chemosensory. More recent studies have reported their expressions in non-neuronal cells such as in synoviocytes, chondrocytes, lung epithelial cells, smooth muscle cells, immune cells, and different types of cancers [19,20]. However, their physiological or pathological functions are not completely understood. Several studies also showed the *TRPA1* and, to a lower extent, *TRPV1* mRNA expressions in keratinocytes, and suggested their potential roles in cell growth, survival, inflammation, and abnormal proliferative processes [21,22,23]. *TRPA1* is highly expressed in pancreatic adenocarcinoma, nasopharyngeal carcinoma, and prostate cancer-associated fibroblast cell cultures [23]. The expression of TRPV1 has been observed in breast cancer [11], prostate carcinoma [24] human pancreatic cancer [10], and tongue squamous cell carcinoma [25,26].

The role of TRPA1 and TRPV1 in tumor genesis is often associated with their effect on cell cycle progression, which is a hallmark of malignant tumors. Cancer progression is linked to the inhibition of pathways leading to apoptosis, which results in uncontrolled cell growth and proliferation. Both these TRP receptors are Ca^2+^-permeable ion channels; they increase intracellular Ca^2+^ concentrations by inducing Ca^2+^ influx through the plasma membrane and releasing the stored Ca^2+^ from the mitochondria and the endoplasmic reticulum. This activates Ca^2+^-dependent signaling pathways such as the mitogen-activated protein kinase/extracellular signal regulated kinase (MAPK/ERK) and phosphatidylinositol 3-kinase/Akt Protein Kinase B pathways. A Ca^2+^ influx into a cancer cell activates calmodulin, leading to the activation of the ERK. This in turn stimulates several regulatory targets in the cytoplasm which have key regulatory roles in cell cycle progression, survival, and nuclear signaling, including MAPK-interacting protein kinase, MAPK-activated protein kinases ribosomal s6 kinase, mitogen- and stress-activated protein kinase, and the protease calpain [27,28].

The results linking the TRPA1 and TRPV1 channel functions with carcinogenesis are controversial. TRPA1 activation in glioma and breast cancer cells induces mitochondrial damage and apoptosis via cytochrome-c, the activation of caspase 3 and 9, and the release of apoptosis-inducing factors and endonuclease G [29,30]. In contrast, other studies suggest that TRPA1 activation promotes cancer growth and metastasis via inducing neovascularization and promoting epithelial cell migration [31]. In addition, lipoyl-derived TRPA1 antagonists have been found to effectively inhibit the migration of osteosarcoma cells and suppress the expression of pro-inflammatory cytokines [8]. Regarding TRPV1, its stimulation suppresses melanoma and colon cancer growth [32,33]. In MG63 human OS cells, besides their TRPV1-dependent mitochondrial dysfunction, ROS overproduction, and activation of c-Jun N-terminal kinase, capsaicin can also induce apoptosis and tumor suppression through the TRPV1-independent AMP-activated protein kinase (AMPK)-p53 pathway [15]. However, the functionalization of TRPV1 had no effect on papilloma growth in mice [28] and the TRPV1 antagonists AMG-980 and SB-705498 did not alter skin carcinogenesis [22,28]. Taking these data together shows that the effect of TRPA1 and TRPV1 activation greatly depends on the cancer cell types and complex sensory–vascular–immune–tumor interactions in the cancer microenvironment.

We found that AITC induced a Ca^2+^ influx in K7M2 mouse OS cells, which was reversed by the antagonist HC-030031, demonstrating its functional TRPA1 expression. However, the concentration-dependent viability-decreasing effect of AITC was not remarkably diminished by the TRPA1 antagonist, suggesting that cell damage, measured by our ATP-based technique, in this system was only partially TRPA1-mediated. This conclusion is supported by the viability-decreasing effect of AITC on CHO cells not expressing the TRPA1 channel and the similar action of AITC in other cancer types 9 HepG2 human hepatocellular carcinoma cells [34] and breast [30], bladder [35], and cervix cancer cells [36]. The capsaicin treatment also induced a significant concentration-dependent decrease in the viability of the K7M2 OS cells. Nevertheless, capsaicin did not have any effect on the Ca^2+^ influx in these cell lines, suggesting its non-TRPV1-dependent action, which was also supported by the low expression of *Trpv1* mRNA, the lack of antagonism by capsazepine, and a similar viability-decreasing effect of capsaicin in CHO cells. These independent pathways may involve mechanisms such as oxidative stress, the enhancement of AMPK and p53, and the activation of pro-apoptotic pathways [37].

The limitation of the present study is that the receptors’ functionality was only investigated in the mouse K7M2 cell line. Since this cell line does not mimic all features of OS, it is difficult to draw strong conclusions about the potential therapeutic applications of TRPA1 agonists. However, since OS is an extremely malignant tumor with therapy resistance and poor prognosis, it can be concluded that the present results are valuable for initiating further research on a broader scale.

## 4. Materials and Methods

### 4.1. Experimental Animals

A total of 9–10-week-old male BALB/c mice were housed in a temperature and humidity-controlled 12 h light–dark cycle at the Laboratory Animal House of the Department of Pharmacology and Pharmacotherapy of the University of Pécs and provided, ad libidum, with standard rodent chow and tap water. All procedures were designed and conducted in full accordance with European legislation (directive 2010/63/EU) and the Hungarian Government’s regulation (40/2013, II. 14.) in reference to the protection of animals used for scientific purposes. The project was approved by the Animal Welfare Committee of the University of Pécs, and the National Scientific Ethical Committee on Animal Experimentation of Hungary (BA02/2000-23/2016). During the experiment all efforts were made to reduce the number of animals used and their suffering.

For the intratibial injection of the K7M2 cells, mice were deeply anesthetized by an intraperitoneal injection of Na- Pentobarbital (Euthanimal, Alfasan Netherland, BV, Woerden, The Netherlands). After shaving and disinfecting one of their hind limbs, a small incision was made on the anterolateral surface under an operating microscope. The tibia was sharply dissected proximally until the metaphyseal flair was identified, where a small hole was created with a 27-gauge needle. The needle was angled st approximately 45° in the sagittal plane and advanced to penetrate the cortex, and then it was twisted gently to create a cortical window but leave the posterior cortex intact, to avoid fracture. K7M2 cells (500,000 cells in 10 μL of PBS) were injected into the defect with a Hamilton pipette. The muscle was pulled back over the bone and the skin was closed in a running fashion using a 4-0 vicryl suture. Control animals received a similar treatment, with the difference being that sterile saline was injected into the bone. Animals were awakened and recovered in the usual fashion.

### 4.2. Human Samples

The human samples were taken during the biopsies of patients with suspected OS. The immunohistochemical experiments were performed after a histopathological confirmation of OS. All procedures were approved by the Regional and Institutional Committee of Science and Research Ethics of the University of Pécs (license No. 9624—PTE 2023).

### 4.3. Cell Lines

The mouse OS K7M2 cell line was provided by Krisztina Buzás, Szeged Biological Research Center, maintained in Dulbecco’s Modified Eagle’s Medium (DMEM, Thermo Fisher Scientific, Waltham, MA, USA) supplemented with 2 mmol L-glutamine and 10% fetal bovine serum (FBS), (Thermo Fisher Scientific) and kept at 37 °C in a 5% CO_2_ incubator.

The CHOK-K1 cell line (ATCC, Manassas, VA, USA) was cultivated in Dulbecco’s Modified Eagle’s Medium (DMEM, Thermo Fisher Scientific, Waltham, MA, USA) supplemented with 4 mmol L-glutamine, 10% fetal bovine serum (FBS), and 1× penicillin/streptomycin (Thermo Fisher Scientific). The cells were maintained at 37 °C within a 5% CO_2_ incubator.

### 4.4. RNA Isolation and PCR Gel Electrophoresis

The K7M2 cell line was homogenized in 1 mL of TRIzol-reagent by vortexing, and its RNA contents were isolated using Direct-zol™ RNA MicroPrep (Zymo Research, Irvine, CA, USA), according to the manufacturer’s instructions. The amount and purity of RNA were determined using a Jenway™ Genova Nano Micro-volume Spectrophotometer (Fisher Scientific, Loughborough, UK). Samples were then treated with 1U DNase I enzyme to eliminate any contaminating genomic DNA. cDNA was synthesized from 500 ng of RNA using an Applied Biosystems™ High-Capacity Reverse Transcription Kit (Thermo Fisher Scientific, Waltham, MA, USA).

PCR was performed using a QuantStudio™ 5 system (Life Technologies Magyarország Ltd., Budapest, Hungary) in a 96-well block, using *Gapdh* as a reference gene, with a reaction volume of 10 µL, containing 1× SensiFAST™ Probe Lo-ROX mix (Meridiane Bioscience, Memphis, TN, USA), 400 nM of probe primer mix (forward and reverse), and 20 ng cDNA. FAM-conjugated TaqMan™ Gene Expression Assays (Thermo Scientific, Waltham, MA, USA) were used to amplify the target loci—*Gapdh*: Mm99999915_g1, *Trpa1*: Mm01227437_m1, and *Trpv1*: Mm01246302_m1. The K7M2 PCR products were electrophoresed on a 2% agarose gel containing 0.01% ethidium bromide at 70 V for 40 min and visualized using a Molecular Imager BioRad Gel Doc XR+ (BioRad Laboratories, Hercules, CA, USA) with Image Lab 6.0.1 build 34 software.

### 4.5. Tissue Collection and Sample Preparation for RNAscope Study

In order to detect the localization/colocalization of *TRPA1/Trpa1* and *TRPV1/Trpv1* in the tissues and cells, the highly sensitive in situ hybridization RNAscope technique was used since their detection at the protein level is technically not possible due to the lack of selective antibodies [38]. We also tested several commercially available and even custom-made antibodies for both TRPA1 and TRPV1 including Abcam (ab58844, ab62053), Aviva (ARP35205_P050), Biorbit (orb86362) and Novus (NB110-40763), but none of them proved to be selective in either tissues or cells in our hands either. Control and tumor-injected mice were deeply anesthetized with Euthasol 14 days after the surgery. A transcardial perfusion was performed with 20 mL of ice-cold 0.1 M PBS, followed by 150 mL of 4% paraformaldehyde solution in Milloning buffer (pH = 7.4). The operated hind limb was removed and post-fixed for 72 h. After decalcification (Morse’s solution for two days), samples were paraffin-embedded and sliced into 5 μm sections. Human samples were post-fixed for a longer time (5 days) and then treated in the same way.

### 4.6. K7M2 Cells’ Preparation for RNAscope

K7M2 cells were centrifuged at room temperature (RT) at 1000× *g* for 5 min. Supernatants were removed and cells were resuspended with media to adjust the cell density to 1 × 10^6^ cells per mL. Adhesive slides with filter cards and sample chambers were assembled in a cytospin centrifuge, then 100 µL of sample was pipetted into each sample chamber. Slides were centrifuged at 1000× *g* for 5 min to allow the cells to form a monolayer on the slide [39]. The RNAscope Technical Note for Cultured Adherent cells protocol (Advanced Cell Diagnostics, Newark, CA, USA, ACD, technical note 320538) was used for the subsequent steps. The cytospin samples were air-dried for 30 min and then fixed using 10% NBF (Sigma) for 30 min at RT. Cells were washed 3 times with 1x PBS and dehydrated in 50%, 70%, and 100% ethanol and stored at −20 °C in preparation for RNAscope staining.

### 4.7. TRPA1/Trpa1 and TRPV1/Trpv1 RNAscope In Situ Hybridization

A RNAscope Multiplex Fluorescent Reagent Kit v2 (ACD, Hayward, CA, USA) was used according to the manufacturer’s protocol. Human and mouse sections were deparaffinized, H_2_O_2_-blocked, boiled, and pre-treated with Protease Plus. K7M2 slides were re-hydrated and treated with Protease III.

All samples were subsequently hybridized with probes specific to human or mouse mRNA—*trpa1* (ACD, Cat. No., 837411-C2,), *trpv1* (ACD, Cat. No., 415381,), ezrin (ACD, Cat. No., 535811)—along with human or mouse 3-plex positive control probes specific to POLR2A mRNA (fluorescein), PPIB mRNA (cyanine 3, Cy3), and UBC mRNA (cyanine 5, Cy5) and 3-plex negative control probes (ACD; Cat. No. 320871) specific to bacterial dabP mRNA. Then, sequential signal amplification and channel development was conducted. Nuclear counterstaining with DAPI was then performed and ProLong Diamond Antifade Mountant was added to the slides for confocal imaging.

Fluorescent images were taken with a Nikon Eclipse Ti2-E confocal microscope (AURO-SCIENCE CONSULTING KFT. Budapest, Hungary) with 20× and 60× magnification. Blue, green, red, and white virtual colors were selected to depict the fluorescent signals of DAPI (nuclear counterstain), Cy3 (*TRPA1/Trpa1* mRNA), fluorescein (*TRPV1/Trpv1* mRNA), and Cy5 (ezrin mRNA), respectively. For K7M2 cells, z-projection (12–15 stacks/image) and a 2 µm interval were chosen. DAPI was excited at 405 nm, Cy3 at 550 nm, fluorescein at 488 nm, and Cy5 at 650 nm. Brightness/contrast were processed using (Fiji, 1.53c, NIH, Bethesda, MD, USA).

### 4.8. Radioactive _45_Ca^2+^ Uptake Experiments on K7M2 Cells

For receptor selectivity experiments, K7M2 cells were investigated in response to AITC (10 µM, 100 µM, 200 µM) as well as capsaicin (100 nM, 1 µM) (Sigma-Aldrich Ltd., Budapest, Hungary) since a concentration higher than 1 µM can cause non-specific effects on receptor-expressing cell lines. These responses were subjected to antagonist treatments by 10 µM HC-030031 (Tocris Bioscience, Bristol, UK, Cat. No. 2896) or capsazepine (Sigma-Aldrich Ltd., Budapest, Hungary), respectively. Cells were seeded in 15 μL of cell culture medium onto Microwell Minitrays (Merck KGaA, Darmstadt, Germany) and incubated overnight at 37 °C in a 5% CO_2_ incubator. The next day, cells were washed and incubated in 10 μL Ca^2+^-free Hank’s solution (pH 7.4) containing capsaicin or AITC and 200 μCi/mL of _45_Ca^2+^ isotope (1.3 Ci/mmole, Ammersham, Buckinghamshire, UK) for 1 min at room temperature. Then, cells were washed with extracellular solution and the remaining buffer was evaporated, the retained isotope was collected in 15 μL of 0.1% SDS, and their radioactivity was measured in 2 mL of scintillation liquid in a Packard Tri-Carb 2800 TR scintillation counter. _45_Ca^2+^ isotope retention is expressed in counts per minute (CPM).

### 4.9. ATP Luminescent K7M2 Cell Viability Assay

Cells were treated with AITC and capsaicin dissolved in DMSO to different concentrations (5 µM, 10 µM, 20 µM, 50 µM, 100 µM, and 200 µM). The DMSO concentrations of the solvent-treated control cells were adjusted to similar concentrations as the concentrations of the treatments (2%, 1%, 0.5%, 0.2%, 0.1% and 0.05%). For the pre-treatment, cells were treated separately with 10 µM HC-030031 and capsazepine for 10 min prior the addition of the agonists AITC and capsaicin, respectively. K7M2 cell viability was assessed using a CellTiter-Glo^®^ Luminescent Cell Viability Assay (CTG, Promega, Madison, WI, USA) mixture as recommended by the manufacturer. Cells were seed in 96-well tissue culture plates at a density of 5000 cells/well in 100 µL of media and cultured for 24 h. On the following day, compounds were added at the above-mentioned concentrations and cells were incubated for 48 h. When this incubation was completed, the plate and its contents were balanced at room temperature for approximately 30 min. Promega CellTiter-Glo^®^ Reagent (Promega, Madison, WI, USA) was added in a volume equal to the cell culture medium present in each well. An ATP-based luminometric measurement of the metabolically active cells in the culture were determined by an EnSpire^®^ Multimode Plate Reader (Waltham, MA, USA) as relative light units (RLUs). Viability was calculated as follows:Viability (%) = RLU of experiment well/RLU of control well × 100.

### 4.10. Statistical Analysis

Statistical analyses were performed using GraphPad Prism 8 software. The distribution of the data was examined and passed the D’Agostino and Pearson normality test, followed by a one-way ANOVA and Dunnett’s post hoc test or Welch’s *t*-test. In all cases, *p* < 0.05 was considered statistically significant.

## Figures and Tables

**Figure 1 ijms-25-03760-f001:**
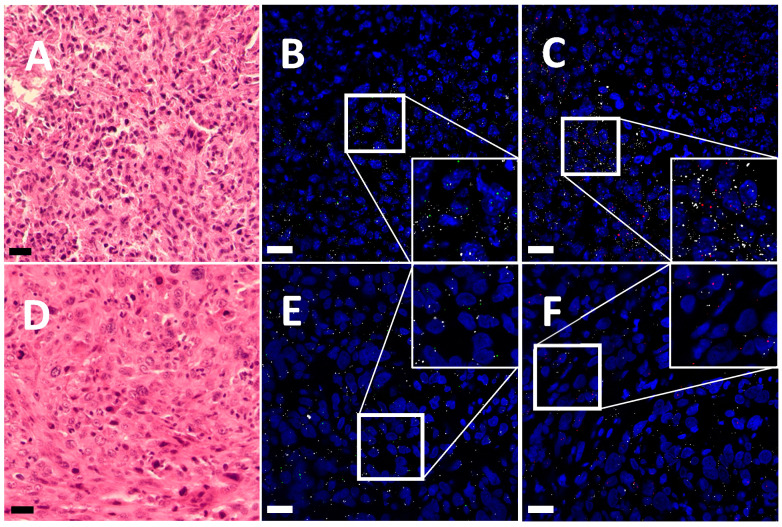
Representative images of *TRPA1* and *TRPV1* expression in OS cells in human tissue. H&E staining of the tissue (**A**,**D**). *TRPA1* mRNA (green), *TPRV1* mRNA (red), and ezrin mRNA (white) are visualized by RNAscope (**B**,**C**,**E**,**F**). Sections are counterstained with 4′,6-diamidino-2-phenylindole (DAPI, blue) for nuclei. n = 2 patients. Scale bars: 25 μm.

**Figure 2 ijms-25-03760-f002:**
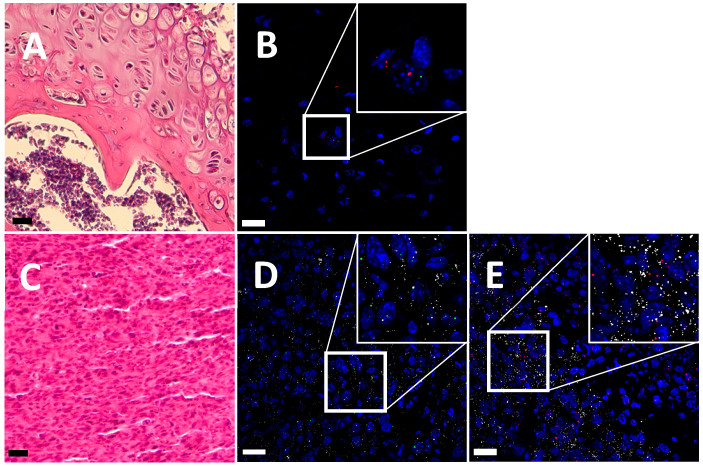
Representative images of *Trpa1* and *Trpv1* expression in mouse tissue. H&E staining of the control (**A**) and OS samples (**C**). *Trpa1* mRNA (green) and *Trpv1* mRNA (red) are detected in the control (**B**) and tumor tissue (**D**,**E**). OS cells are identified by their ezrin mRNA (white). Sections are counterstained with DAPI (blue) for nuclei. n = 2 control mice and 3 tumor mice. Scale bars: 25 μm.

**Figure 3 ijms-25-03760-f003:**
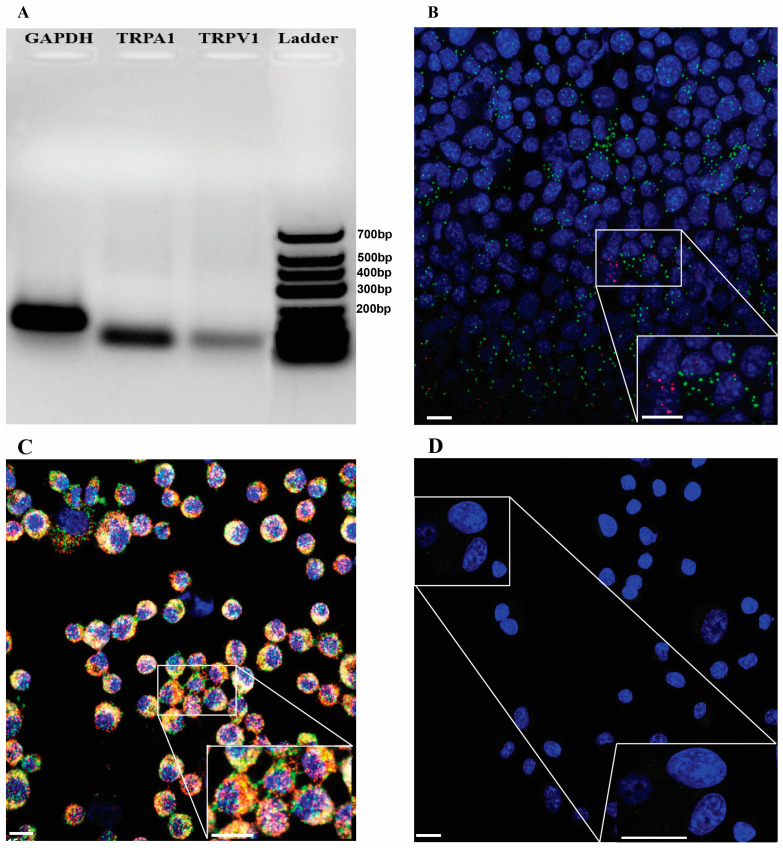
Representative images of gel electrophoresis of glyceraldehyde 3-phosphate dehydrogenase (*Gapdh)*, *Trpa1*, and *Trpv1* (**A**), RNAscope showing expression of *Trpa1* mRNA (green) and *Trpv1* mRNA (red) in OS K7M2 cells (**B**), and positive and negative controls (**C**,**D**). RNAscopes were depicted and counterstained with DAPI (blue) for nuclei. RNAscope images were captured from slides containing 100 µL of a cell suspension at a concentration of 1 × 10^6^ cells/mL. Scale bars: 15 μm.

**Figure 4 ijms-25-03760-f004:**
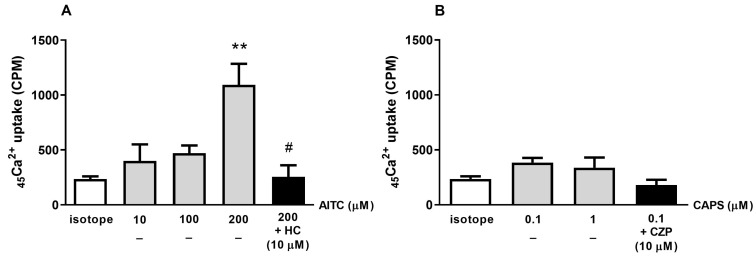
Effects of the TRPA1 agonist AITC (**A**) and the TRPV1 agonist capsaicin (CAPS) (**B**) on the _45_Ca^2+^ uptake in K7M2 cells. Pretreatments with the TRPA1 antagonist HC-030031 (HC) (**A**) and the TRPV1 antagonist capsazepine (CZP) (**B**) reverse the effects of AITC and capsaicin, respectively. Each column represents the mean ± SEM of n = 3 independent experiments (biological replicates), each in technical triplicates. ** *p* < 0.01 (vs. control; one-way ANOVA, Dunnett’s post hoc test); # *p* < 0.05 (vs. 200 µM AITC; one-way ANOVA, Dunnett’s post hoc test).

**Figure 5 ijms-25-03760-f005:**
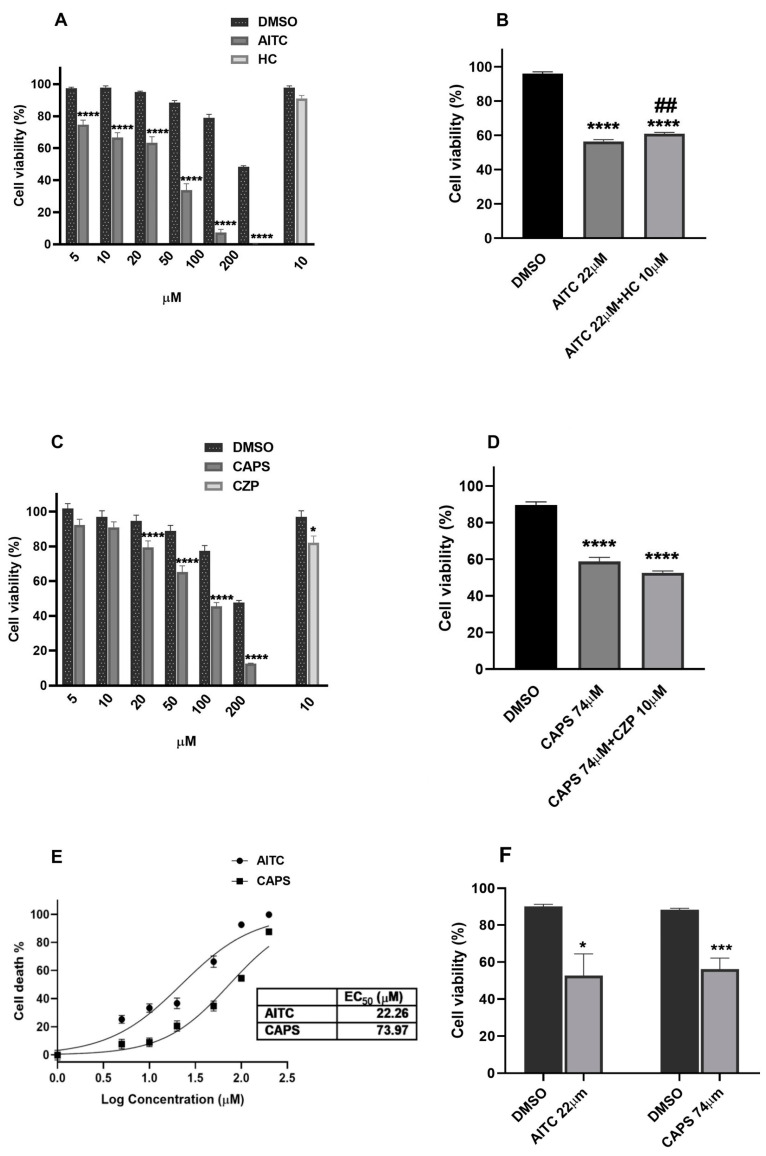
Effects of the TRPA1 agonist AITC and the TRPV1 agonist capsaicin (CAPS) on the viability of K7M2 cells. Treatment with AITC or capsaicin causes a concentration-dependent reduction in K7M2 viability (**A**,**C**). Pretreatment with the TRPA1 antagonist HC-030031 (HC) blocks AITC’s effect (**B**). Pretreatment with the TRPV1 antagonist capsazepine (CZP) has no effect on capsaicin-induced cell death (**D**). Dose–response curve of the effect of AITC and capsaicin on cell death, with EC_50_ values of 22 µM and 74 µM, respectively, (**E**). AITC and capsaicin decrease the viability of CHO cells not expressing TRPA1 or TRPV1 receptors (**F**). All data are expressed in mean ± SEM of n = 3 experiments (biological replicates), each in technical triplicates. **** *p* < 0.0001, *** *p* < 0.001, * *p* < 0.05 (vs. DMSO; one-way ANOVA with Dunnett’s post hoc test or Welch’s *t*-test), ## < 0.01 (vs. 22 µM AITC; one-way ANOVA with Dunnett’s post hoc test).

## Data Availability

The data presented in this study are available on request from the corresponding author.

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
