# Peer review of "Transient Receptor Potential Ankyrin 1 Ion Channel Is Expressed in Osteosarcoma and Its Activation Reduces Viability"

_ijms, 2024, doi:10.3390/ijms25073760_

Round 1
Reviewer 1 Report (Previous Reviewer 2)
Comments and Suggestions for Authors
The paper entitled “Transient Receptor Potential Ankyrin 1 Ion Channel Is Expressed in Osteosarcoma and Its Activation Reduces Viability” is presented. The study is potentially interesting. The methods used were appropriate and the conclusions were reasonable.
There are some issues
Figure 1 Can you state the samples size of analysis (n=?) in the figure legends, and quantitative measurement ? Can you confirm some of these results by qPCR or western blots?
Figure 2 Can you state the samples size of analysis (n=?) in the figure legends, and quantitative measurement ? Can you confirm some of these results by qPCR or western blots?
Figure 3 Can you state the samples size of analysis (n=?) in the figure legends, and quantitative measurement ? Can you confirm some of these results by qPCR or western blots?
It was mentioned that this paper aims to characterize the expression and functionality of TRPA1 and TRPV1 channels in human and mouse osteosarcoma tissues and in mouse cell line. Many studies have found by scRNAseq analyses (for example PMID: 37106386; PMID: 37439349) the Heterogeneity of osteosarcoma tissues. It would be informative to discuss if TRPA1 and TRPV1 are expressed in these cell types.
Author Response
Please see the attachment

Reviewer 2 Report (New Reviewer)
Comments and Suggestions for Authors
In the paper entitled “Transient Receptor Potential Ankyrin 1 Ion Channel Is Expressed in Osteosarcoma and Its Activation Reduces Viability” by Hudhud and coworkers, it is investigated the expression of TRPA1 and TRPV1 receptors and their role in osteosarcoma.
Even though TRPA1 was already investigated in this tumor type, Authors characterize the expression of TRPV1 receptor. The study is interesting but in my opinion could be implemented with few experiments.
Comments on the paper:
Abstract
I would not specify amounts of agonist/antagonist used. It is something that it is preferably to be specified in the M&M or Results sections.
I would substitute keywords like RNA scope or cell viability with more specific ones.
Introduction
Line 59: I’m not sure OS patients are usually treated with radiotherapy.
Results
Line 128: It is not specified which cell line was used as positive/negative control. Could Authors justifiy the so different intensity signal? The K7M2 cells seem to express very low levels of these channels, especially TRPV1.
Line 133-136: Why did the Authors performed the assay with different agonists concentrations while they chose just one concentration of antagonist compounds? I would suggest to perform the assay with different concentrations of antagonist.
Line 136: Why did the Authors used different concentrations of agonists for the channels? Why did they stop testing the TRPV1 agonist at 1 uM? I would suggest to use the same concentrations as the TRPA1 agonist (higher than those used) otherwise justify why they chose different concentrations.
FIGURE 5:
Indicate how many times the experiments were performed. It looks like the graph depict technical replicates rather than biological. It misses the description of the (F) graph in the figure legend.
Figure 5 F: How could the authors justify the decrease in CHO cell viability if they do not express the channels?
Just correct some typos in the text.
Author Response
Please see the attachment

Reviewer 3 Report (New Reviewer)
Comments and Suggestions for Authors
no
Author Response
Please see the attachment.

Reviewer 4 Report (New Reviewer)
Comments and Suggestions for Authors
Dear Authors,
please find my comments in the attached file.

None
Round 2
Reviewer 2 Report (New Reviewer)
Comments and Suggestions for Authors
The Authors addressed all my questions. Thank you.
Just a few text editing.
For example at line 364 it misses "uM" referred to one of the AITC concentrations.
Just read carefully the manuscript before publishing to solve any other typos or any missing information.
Reviewer 3 Report (New Reviewer)
Comments and Suggestions for Authors
The authors have addressed the previous questions in the new version, no more new questions.
This manuscript is a resubmission of an earlier submission. The following is a list of the peer review reports and author responses from that submission.
Round 1
Reviewer 1 Report
Comments and Suggestions for Authors
In the submitted manuscript, the authors aim to show that the two receptors TRPA1 and TRPV1 are functionally expressed in human osteosarcoma. For documenting the expression, they use the little established method RNAscope. The images shown in Figure 3 indicate that both receptors are likely to be expressed in the mouse osteosarcoma cell line used, with TRPA1 providing significantly more and stronger signals in the selected section. However, compared to the positive control, one can probably only speak of a weak expression Also in the murine tissue, few cells in the normal and malignant tissue section show a weak expression signal. These expression data are not very clear and should be supplemented by immunostaining. Oguri et al (PMC7797518) have already published this method successfully with this very receptors and working antibodies should not be the limiting problem.
In the next experiments, the authors use recognized agonists of the two receptors and show that calcium uptake is increased by the presence of the TRPA1 agonist AITC, at least at the 200 µM concentration. I would like to have revised the authors' statement that the TRPV1 agonist capsazepine also causes a slight effect on Ca uptake, since a statistically non-significant change is by definition not existing. The antagonist of the TRPA1 receptor H-030031 is able to abolish the effect of the agonist. In the last part of the experiment, the authors show that the agonists of both receptors reduce the viability of the cells under investigation. This effect is hardly or not influenced by the antagonists. A control with cells not expressing the receptors need to be included to exclude an experimental artefact. Furthermore, the results should be replicated in other cell lines. It should indicate how many replicates were made in how many independent experiments.
In summary, the submitted manuscript is at most a preliminary draft and the key message that TRPA1 is expressed in osteosarcomas and that agonists of this receptor induce cell death has already been published (PMC9667541).
Comments on the Quality of English LanguageThe quality of English Language is appropriate
Reviewer 2 Report
Comments and Suggestions for Authors
The paper entitled Transient Receptor Potential Ankyrin 1 Ion Channel Is Expressed inOsteosarcoma and Its Activation Reduces Viability is presented. The study is potentially interesting. The methods used were appropriate and the conclusions were justified.
There are some issues
Figure 1. Representative images of TRPA1 and TRPV1 expression on OS cells in human tissue. Can you state the sample size of analyses, n=? in the figure legends? Can you provide quantitative measurements?
Figure 2, 3, Can you state the sample size of analyses, n=? in the figure legends? Can you provide quantitative measurements?
Figure 4, 5 Can you state the sample size of analyses, n=? in the figure legends?
Regarding the effects of the TRPA1 agonist AITC (A) and the TRPV1 agonist capsaicin (CAPS), can you use overexpression and knockdown expression of TRPA1 and TRPV1??
Many studies have found by single-cell RNAseq analyses (for example PMID: 34367994; PMID: 37439349; PMID: 37034356 ) osteosarcoma tissues contain many cell types. It would be informative to discuss if TRPA1 and TRPV1 are expressed in these cell types.